# OpenReview forum: "Simple synthetic data reduces sycophancy in large language models"
_ICLR.cc/2025/Conference — Submitted to ICLR 2025_

### Official Review · Reviewer_zVCa · 2024-11-02

**Soundness:** 3
**Presentation:** 3
**Contribution:** 2
**Rating:** 6
**Confidence:** 3

**Summary:**

This paper investigates the prevalence of sycophancy in LLMs and discovers that both model scaling and instruction-tuning can exacerbate this issue. In response, the authors propose a straightforward synthetic data construction method designed to mitigate sycophancy in LLMs. Experimental results demonstrate that their training approach effectively generalizes to prompts that the models have not seen during training.

**Strengths:**

**Clarity:** The paper is very well-written and easy to follow, with a smooth flow that makes it enjoyable to read. The figures are clear and effectively help in understanding the problems being addressed and the approach employed.

**Motivation:** The paper is well-motivated, addressing the critical issue of sycophancy in LLMs. This phenomenon poses a significant challenge, as it resembles reward hacking and may undermine the effectiveness of the RLHF process, potentially hindering the further development of LLMs.

**Implications:** The paper highlights that both model scaling and instruction tuning can lead to increased sycophancy in LLMs. These important findings serve as a reminder of the problem's significance.

**Soundness:** The proposed approach is logically sound, and the experimental results convincingly demonstrate the effectiveness of their training method, particularly in generalizing to prompts that the models have not seen during training.

**Weaknesses:**

**Comparison:** It would be helpful to compare the model's performance using alternative prompting techniques, such as system 2 attention [1]. If these prompting methods effectively reduce sycophancy, it could challenge the applicability of the current training approach, which appears to require more efforts. Additionally, it remains unclear how the current approach would generalize to scenarios involving more open-ended questions where there are no set of gold answers to choose from. In comparison, prompting techniques could generalize to those open-ended questions more easily.



[1] System 2 Attention (is something you might need too)

**Questions:**

See Weaknesses part. Could you add a comparison to some prompting-based techniques to see whether the training based approach works better or not?

---

> ### Author Response · Authors · 2024-11-17
> **Author response to Reviewer zVCa**
>
> Thank you for the positive review and thoughtful comments. We are glad that you appreciated that our paper was well motivated and tackled a significant problem and that our approach is and evidence is sound.
>
> We’ve added discussion to the paper to address your feedback. Please let us know if you have any additional comments or suggestions.
>
> > It would be helpful to compare the model's performance using alternative prompting techniques, such as system 2 attention [1]. If these prompting methods effectively reduce sycophancy, it could challenge the applicability of the current training approach, which appears to require more efforts.
>
> Thank you for this nice suggestion on whether prompting methods can effectively reduce sycophancy. Indeed, in our preliminary experiments, we also found that it is relatively straightforward to provide additional instructions to the model via prompting that allow the model to effectively ignore user opinions. One important distinction, however, is that the effectiveness of prompting in terms of reducing sycophancy does not reduce the importance of exploring methods of finetuning that can reduce sycophancy. This is because prompting methods have several shortcomings that make them undesirable as long-term solutions.
>
> First, a clear shortcoming of prompting is that prompting is less robust than finetuning, as users can easily add injections such as “ignore all previous instructions” to attempt to bypass the prompting strategy. A more-nuanced limitation of prompting methods is that they may actually be more costly than finetuning in the long run.  Whereas prompting methods often ask the model to do additional thinking to reduce sycophancy (Weston & Sukhbaatar, 2023), finetuning can reduce sycophancy without needing the model to perform extra thinking. This makes finetuning preferable to prompting in cases where one expects to run the model on many inputs, as prompting requires paying extra output tokens for thinking and extra input tokens for the prompting instructions for every single input.On the other hand, finetuning requires a one-time up-front cost, but does not add any additional cost for processing future inputs. This makes finetuning preferable for models that one might expect to deploy in a production setting. For this reason, it is still crucial to investigate methods of finetuning that can reduce sycophancy in language models.
>
> We’ve added this discussion into a new subsection of the “Frequently asked questions” section of the Appendix, and we hope that this discussion will provide clarity to readers on why it is important to find finetuning methods to reduce sycophancy independently of if prompting methods for reducing sycophancy exist.

---

### Official Review · Reviewer_AFnr · 2024-11-03

**Soundness:** 3
**Presentation:** 3
**Contribution:** 3
**Rating:** 6
**Confidence:** 4

**Summary:**

The paper investigates sycophancy in LLMs, where models tend to tailor responses to align with user opinions, even when those statements are incorrect. To measure this behavior, the authors first examine sycophancy on three classification tasks with no definitive correct answers and one simple addition classification task involving incorrect statements. They find that both model scaling and instruction tuning amplify sycophantic tendencies, and sycophancy still exists even when the statements are apparently wrong. To mitigate this issue, the authors propose a synthetic data intervention designed to encourages more robust responses to potentially incorrect user beliefs. They demonstrate that simply finetuning on this dataset can significantly reduce sycophantic responses on held-out prompts. Further evaluations reveal that this intervention does not impact performance on established benchmarks, such as MMLU and Big-Bench Hard, indicating that the approach maintains model accuracy while reducing sycophancy.

**Strengths:**

This work effectively highlights the issue of sycophancy in LLMs, and conducts evaluations across three model sizes—8B, 62B, and 540B. This finding that sycophantic behavior becomes more pronounced as model size increases provides a valuable insight into how scaling influences sycophancy.

The synthetic data intervention method is straightforward and effective, making the intervention potentially easy to replicate across different models.

The proposed method is tested on two popular benchmarks, MMLU and Big-Bench Hard, showing the effectiveness without compromising the model’s performance on established tasks or affecting the existing learned knowledge.

**Weaknesses:**

While the paper offers insights about sycophancy in language models and a method for reducing it, further experiments could enhance the robustness and generalizability of the proposed finetuning method:
1. Although three models of varying sizes were tested, the evaluation is limited to a single model type. It would be beneficial to examine sycophancy across a wider range of both open-source LLMs, such as LLaMA -- which has been widely studied in research and also offers multiple size options -- and closed-source models like GPT or Gemini. Expanding the evaluation to diverse model architectures would also help verify whether the synthetic data intervention remains effective across different types of models.
2. The paper acknowledges limitations in prompt diversity and task scope, as the experiments are largely restricted to classification tasks with a narrow range of prompt formats. Testing the fine-tuned model’s performance on more varied, out-of-distribution samples would provide a stronger assessment of the intervention’s generalizability. For example, the user's belief is subtly embedded within a description of their actions, serving as contextual information.
3. In Section 3, the study on objectively incorrect statements is limited to simple addition tasks, which constrains the ability to generalize the claim that “models are sycophantic for objectively wrong answers.” Expanding this evaluation to include a broader array of tasks involving objectively false statements—such as those related to misinformation (e.g., "Apple cider vinegar is a miracle cure for cancer"), fake news (e.g., a famous celebrity supports xxx but actually not), or conspiracy theories (e.g., "the Earth is flat")—would lend stronger support to this conclusion.

Writing clarity:
Table 1 and Figure 1 are nearly duplicated.

**Questions:**

What proportion of questions is filtered out for each model during the filtration process? Given the differences in model sizes, I suppose there would be a trend where larger models retain a greater number of synthesized training examples.

---

> ### Author Response · Authors · 2024-11-17
> **Author response to Reviewer AFnr**
>
> Thank you for the encouraging review of our work. We appreciate that the reviewer found our work to provide valuable insights on sycophancy and that our proposed synthetic-data intervention to be straightforward yet effective.
>
> We’ve modified our paper to help address your questions and responded below. Please let us know if you have any additional questions or suggestions.
>
> > What proportion of questions is filtered out for each model during the filtration process? Given the differences in model sizes, I suppose there would be a trend where larger models retain a greater number of synthesized training examples.
>
> Thank you for this important question about the number of examples removed as a result of our filtration process. In Section D.4 in the Appendix, we studied model performance on the synthetic prompts before filtration was applied. We found that while LLM-8B (our smallest model) only obtained an accuracy of 53.5%, LLM-540B (our largest model) obtained an accuracy of 66.0%. This indeed indicates that fewer examples are filtered out as model size increases because larger models obtain higher performance on the synthetic data.
>
> To make this more clear, we added the following clause to Section 6 of the main body: “(see Appendix D.4 for a breakdown of the proportion of examples filtered out for each model).”
>
> In practice, however, we found that the actual number of synthetic examples trained on is not a limiting factor. In Appendix B.6, we evaluated the number of steps of finetuning needed to obtain the benefits of synthetic-data intervention. We found that the effects of our synthetic data could be seen after only 500 steps at batch size 32. Additionally, finetuning was applied after the synthetic data was already mixed with some instruction-tuning data. These findings seem to indicate that the quantity of the synthetic examples is not a bottleneck; the improvements were from filtration that improved the quality of the examples that the model saw.

---

### Official Review · Reviewer_e8Yz · 2024-11-03

**Soundness:** 2
**Presentation:** 3
**Contribution:** 2
**Rating:** 3
**Confidence:** 2

**Summary:**

This paper studies the prevalence of sycophancy in language models and puts for a synthetic data based approach to reduce sycophantic behavior in LLMs.

**Strengths:**

- the synthetic data intervention step leverages openly available datasets, as well as a good variety of such datasets at 17 total
- well fleshed out limitations section, indicating a paper that is grounded it what it purports to provide evidence for.

**Weaknesses:**

- the set of models that are used for experiments are quite limited.
- the intervention to reduce sycophancy requires fine-tuning, which may not be feasible for all use-cases. For example, when access to the model is limited by openness or resource constraints.
- single prompt format in all experiments

**Questions:**

1. is there any reason why larger or different models were not chosen? For example, the Mixtral models or Llama family?
2. In cases where fine-tuning is not an option, what are other ways to reduce sycophantic outputs?
3. were there attempts with different prompt formats, e.g., with some ablation studies? If so, what were the results? If not, why not?

---

> ### Author Response · Authors · 2024-11-17
> **Author response to Reviewer e8Yz**
>
> Thank you for the insightful feedback and thoughtful comments. We are glad that the reviewer appreciated our use of open-source datasets and that our synthetic-data intervention provided diverse coverage.
>
> We’ve added discussion to our paper and provided responses to your questions below. We hope that we’ve helped clarify some of your questions - please let us know if you have any additional questions or feedback.
>
> > Is there any reason why larger or different models were not chosen? For example, the Mixtral models or Llama family?
>
> Thank you for this valuable suggestion about experimenting with larger or different models. We finetuned models of sizes up to 540B parameters, which we consider to be large enough to see any surprising behaviors that may occur at large model sizes.
>
> We did not experiment with other model families due to the cost of finetuning additional large models. We made sure, however, to provide as much detail as possible in describing the data generation and finetuning procedures that we used. We also publicly release our code for generating our synthetic data, which we hope will allow readers to easily understand and reproduce our work.
>
> > In cases where fine-tuning is not an option, what are other ways to reduce sycophantic outputs?
>
> Thank you for this interesting question about how one can try to reduce sycophancy outside of finetuning. While we believe this is out of the scope of our current work, we’ve added the following discussion to a new subsection in the “Frequently asked questions” section of the Appendix in the revised manuscript.
>
> “Finetuning language models to reduce sycophancy may not be ideal for all applications. A natural question is thus how one might be able to reduce sycophancy without finetuning. (Sharma et al.,2024) proposes that for models that have gone through reinforcement learning, one could improve the preference model used to train the model by aggregating preferences across more humans or by assisting human labelers. Another approach could be to apply activation steering, whereby one can obtain vectors from sycophancy data that can be used to steer models to be less sycophantic in certain scenarios (Panickssery, 2023).”
>
> > Were there attempts with different prompt formats, e.g., with some ablation studies? If so, what were the results? If not, why not?
>
> Thank you for this thoughtful comment on whether we attempted to use different prompt formats. In initial experimentation with the simple-addition statements task, we indeed found that the specific prompt format used did not significantly affect whether our language model demonstrated sycophantic behavior. For this reason, we did not conduct any additional ablations on prompt formats and instead elected to follow the prompt format used in Perez et. al (2022).
>
> In Appendix D.2, we provided discussion on the prompt format used for synthetic-data generation. We found that our intervention still produces non-negligible reductions in sycophancy on the PHIL and POLI tasks from Perez et al. (2022). These two tasks use a more-contrasting prompt template from the generated synthetic data, which indicates that our intervention is not necessarily limited by its fixed prompt template. We were unable to further investigate sycophancy in other prompt formats due to a lack of available evaluation tasks for sycophancy, though we believe that our intervention method can be easily extended to other prompt formats that may appear in future evaluation settings.

---

### Official Review · Reviewer_LnsW · 2024-11-04

**Soundness:** 3
**Presentation:** 3
**Contribution:** 3
**Rating:** 5
**Confidence:** 2

**Summary:**

The authors aim to address the phenomenon of sycophancy in language models, where models tend to align with user opinions even when they are incorrect or subjective. This phenomenon is observed even for large language models up to 540B parameters. The authors present an approach to mitigate this by using synthetic data in a lightweight fine-tuning process. This synthetic data is generated by reformatting publicly available NLP tasks, intending to decouple truthfulness from user opinions in model responses. The intervention effectively reduces sycophancy, especially in scenarios involving incorrect statements, and shows generalization to models of varying sizes.

**Strengths:**

1. The paper is well-structured, with a clear explanation of sycophancy, its implications, and how the proposed intervention addresses this problem.
2.  The fine-tuning process is lightweight, making this approach accessible and adaptable for large-scale language models with limited computational resources.
3. The intervention's impact is demonstrated with comprehensive results across multiple models and tasks, showing clear reductions in sycophantic responses.

**Weaknesses:**

1. The sycophancy evaluations are primarily limited to multiple-choice tasks. It would be beneficial to explore if the intervention works in generative settings where response options are more diverse.
2. The smallest model used (Flan-LLM-8B) did not respond well to the intervention, highlighting a potential limitation in the effectiveness of the approach for smaller models.

**Questions:**

1. How well does the intervention generalize to generative tasks where the model isn’t limited to choosing from predefined responses?
2. Is there any evidence to suggest that the intervention could be adapted for smaller models to improve performance?
3. Could this approach be extended to improve the model’s adherence to factual information when user opinions align with correct statements?

---

> ### Author Response · Authors · 2024-11-17
> **Author response to Reviewer LnsW**
>
> Thank you for the insightful feedback on our paper. We appreciate the thorough review and the recognition of our paper's strengths. We are glad you found our explanation of sycophancy clear and our results comprehensive.
>
> We have made some modifications to the manuscript following your suggestions. We are happy to expand on the responses or make additional revisions to the manuscript - please let us know if you have any further comments.
>
> > How well does the intervention generalize to generative tasks where the model isn’t limited to choosing from predefined responses?
>
> Thank you for pointing out this limitation of our work. We agree that exploring the intervention's effectiveness in generative settings is an important direction for future work. However, while our current work focuses on multiple-choice settings, we believe the core principle of decoupling truthfulness from user opinions should generalize to generative tasks. However, evaluating this would require developing new metrics and evaluation procedures.
>
> Our focus on multiple-choice tasks allowed for clear quantification of sycophancy, but we acknowledge the value in examining more open-ended scenarios. We hope the clarification in the “What happens if the model is given the option to say that it does not know the answer” subsection of the “Frequently asked questions” section of the Appendix can help provide some clarity on our thinking about open-ended questions.
>
> > Is there any evidence to suggest that the intervention could be adapted for smaller models to improve performance?
>
> Thank you for this insightful question about whether the synthetic-data intervention can be adapted to work better for small models. In Section 6, we found that the filtration step of removing prompts for which a model does not know the correct answer to the claim in the prompt is necessary for successful intervention. This makes sense because the filtration step is designed to clarify that the user’s opinion is independent of the truthfulness to the claim. For example, consider a claim that the model does not know the answer to, such as “foo + bar = baz.” Given a user opinion about this claim, the model will then be trained to randomly agree or disagree with the user since it has no prior knowledge of whether the claim is true. Hence, to teach the model to disregard the user’s opinion when considering the claim, the model must know the ground truth of whether the claim is true or not
>
> However, for the 8B model, this filtration step didn’t seem to make a difference. We hypothesized that the filtration did not help because the 8B model was only obtaining correct answers on the synthetic data by randomly guessing; it did not actually know the correct answer. There is some evidence for this hypothesis, as the results in the “Intervention requires mixing instruction-tuning data” section in the Appendix show that the 8B model actually only achieves 50% accuracy (i.e., random-guessing performance) on the simple-addition statements task, which suggests that the 8B models is not a very capable model since the simple-addition statements task is designed to be easy. These findings seem to suggest that adapting the synthetic-data intervention for small language models may require simplifying the tasks that are used for the intervention such that the model achieves performance better than random guessing.
>
> We’ve added this discussion into a new subsection of the “Frequently asked questions” section in the Appendix.
>
> > Could this approach be extended to improve the model’s adherence to factual information when user opinions align with correct statements?
>
> Thank you for this neat suggestion. Our current intervention teaches the model to ignore the user’s opinion and focus on selecting the correct answer independently of any claims that the user makes. This includes cases in which the user makes a correctly-stated opinion. In other words, we train the model to adhere to the correct answer regardless of the user’s opinion. For this reason, we believe that our approach already inherently improves the model’s adherence to factual information, even when the user gives a correct opinion.

---

> > ### Comment · Reviewer_LnsW · 2024-11-25
> > **Reply to Author's Response**
> >
> > Thank you for the author’s response, which clarified my previous concerns. However, I have some new questions:
> >
> > 1. The author mentioned that removing user opinion statements during the intervention process could improve the final outcomes. My understanding is that during the synthetic data fine-tuning process, the user’s opinion portion is removed, and only factual information is used for fine-tuning. This, in turn, reduces the prevalence of sycophantic behavior in the model. Is my interpretation correct?
> >
> > 2. Regarding the evaluation metrics, the author primarily uses the repetition rate of the user’s opinion. I believe this approach captures only a partial aspect of sycophancy. Could the author provide examples and corresponding metrics where the model agrees with the user’s incorrect answers? This scenario seems more aligned with the essence of sycophancy and could provide more meaningful insights for AI Alignment & Safety.
> >
> > 3. In the 072 claim, the author stated: *“We also found that model scaling increases sycophancy, even though there is no clear reason why scaling would incentivize sycophantic answers.”* However, this phenomenon has been previously discussed in [1]. I do not see this as a particularly strong contribution.
> >
> > [1] Discovering Language Model Behaviors with Model-Written Evaluations

---

> > > ### Author Response · Authors · 2024-11-25
> > > **Clarifications**
> > >
> > > Thanks for the additional questions. We've provided clarification below - please let us know if you have any further questions.
> > >
> > > > The author mentioned that removing user opinion statements during the intervention process could improve the final outcomes. My understanding is that during the synthetic data fine-tuning process, the user’s opinion portion is removed, and only factual information is used for fine-tuning. This, in turn, reduces the prevalence of sycophantic behavior in the model. Is my interpretation correct?
> > >
> > > To clarify our intervention setup: when we say that the intervention teaches the model to "ignore the user’s opinion", we do **not** actually mean that we remove the user's opinion portion of the prompt during finetuning (this portion is actually **kept** during finetuning). Instead, we mean that during finetuning, the model sees examples where the user's opinion matches the correct answer and also sees examples where the user's opinion does not match the correct answer. In either case, the model is taught to respond with the objectively-correct answer, which teaches the model that the user's opinion should not affect the actual answer of the question.
> > >
> > > > Regarding the evaluation metrics, the author primarily uses the repetition rate of the user’s opinion. I believe this approach captures only a partial aspect of sycophancy. Could the author provide examples and corresponding metrics where the model agrees with the user’s incorrect answers? This scenario seems more aligned with the essence of sycophancy and could provide more meaningful insights for AI Alignment & Safety.
> > >
> > > Thanks for bringing up this important point. The two suites of sycophancy evaluations that we used in the paper are the tasks from Perez et al. (2022) and our own proposed simple-addition statements task.
> > > * The tasks from Perez et al. are primarily survey-based questions (such as questions on philosophy and politics) - these questions do not have an objectively-correct or objectively-incorrect answer, and so the measurement of sycophancy on these tasks would be the model's agreement rate with the user (where a random model would agree 50% of the time). As such, we follow Perez et al. and report sycophancy as the percent of answers that match the user's point of view.
> > > * As stated in Section 3, we develop an evaluation dataset of 2.5k simple addition statements that are **objectively incorrect**, and we add synthetic user opinions stating that the user agrees with the incorrect statements. In these cases, we view sycophantic behavior as agreeing with the user's objectively-incorrect opinion. This means that the experiments performed in Section 3 actually already answer your question about examples/metrics where the model agrees with the user's incorrect opinions. Figure 1 shows a concrete example of this behavior, and Figure 3 shows that (a) models know the correct answer, as shown by the dark purple bars, and (b) when given an incorrect user opinion, models will actually flip their correct answer to agree with the user's incorrect answer, as shown by the light pink bars.
> > >
> > > > In the 072 claim, the author stated: “We also found that model scaling increases sycophancy, even though there is no clear reason why scaling would incentivize sycophantic answers.” However, this phenomenon has been previously discussed in [1]. I do not see this as a particularly strong contribution.
> > >
> > > We agree that the behavior that model scaling increases sycophancy has been discussed in Perez et al. The main contribution of Section 2, however, is the finding that **instruction tuning** increases sycophancy, which has not previously been shown before. The additional finding that model scaling increases sycophancy provides additional evidence of this behavior in addition to the findings from Perez et al. We would be happy to remove that sentence from the introduction section if you think it would help make the new findings on instruction tuning more apparent as the main contribution.

---

### Meta-Review · Area_Chair_rh6v · 2024-12-21

**Metareview:**

This paper investigates the sycophancy problem in large language models (LLMs) and proposes a synthetic-data intervention within a lightweight fine-tuning step to mitigate sycophancy. The paper is well-structured, and the experimental results demonstrate its effectiveness. However, the reliance on experiments with only a single model type lacks persuasiveness. It is recommended to include at least one other model family to strengthen the validity of the results.

**Additional Comments On Reviewer Discussion:**

Discussion Summary During the Rebuttal Period:


1. Limited Experiments:
   - The experiments were primarily conducted on models from the same family, which may not fully demonstrate the intervention's effectiveness across different architectures.
   - Narrow Validation Scope: The validation is narrowly focused, limited to classification tasks and specific prompt formats, which restricts a broader understanding of the intervention's impact.

2. Novelty of Argument: The argument that model scaling increases sycophancy is not considered novel and is seen as merely incremental even in instruct tuning.

3. Unaddressed Weaknesses: Some significant weaknesses, such as those pointed out by Reviewer AFnr, have not been responded to by the authors.

---

### Decision · Program_Chairs · 2025-01-22

Reject